# Metabolic Profiles Reveal Changes in the Leaves and Roots of Rapeseed (*Brassica napus* L.) Seedlings under Nitrogen Deficiency

**DOI:** 10.3390/ijms23105784

**Published:** 2022-05-21

**Authors:** Xinjie Shen, Ling Yang, Peipei Han, Chiming Gu, Yinshui Li, Xing Liao, Lu Qin

**Affiliations:** 1Oil Crops Research Institute of Chinese Academy of Agricultural Sciences, Key Laboratory of Biology and Genetics Improvement of Oil Crops, Ministry of Agriculture and Rural Affairs, Wuhan 430062, China; ylssxj@163.com (X.S.); m15665087737@163.com (L.Y.); guchiming@foxmail.com (C.G.); liyinshui@caas.cn (Y.L.); 2Agricultural College, Guizhou University, Guiyang 550025, China; 3Institute of Agriculture Science in Jiangsu Coastal Area, Yancheng 224002, China; hanpeipei_123@126.com

**Keywords:** rapeseed, nitrogen deficiency, root architecture, metabolome, plant hormones

## Abstract

Rapeseed (*Brassica napus* L.) is an important oil crop species and plays a crucial role in supplying edible oil worldwide. However, rapeseed production in the field is often severely inhibited due to nitrogen (N) deficiency. Metabolites play key roles in plant growth and resistance to environmental stress, but little is known about the differential synthesis and accumulation of metabolites underlying rapeseed adaptation to N deficiency. Here, we studied the phenotypic response and used LC–electrospray ionization (ESI), ESI–MS/MS, and widely untargeted metabolomic approaches to detect differences in rapeseed under normal N (HN) and N-deficient (LN) conditions. The results showed that N deficiency severely inhibited rapeseed shoot growth and promoted rapeseed root architectural changes under LN conditions. In total, 574 metabolites were detected, and there were 175 and 166 differentially accumulated metabolites in the leaves and roots between the HN and LN conditions, respectively. The significantly differentially accumulated metabolites were involved in four primary metabolic pathways, namely, sucrose, phenylalanine, amino acid, and tricarboxylic acid cycle metabolism. Notably, we found that plant hormones have distinct accumulation patterns in rapeseed and coordinate to play crucial roles in both maintaining growth and protecting against damage from plant disease under HN and LN conditions. Moreover, our results indicated that flavonoid compounds, especially anthocyanins and rutin, may play important roles in increasing root cell resistance to oxidative damage and soil pathogen infections. Overall, this work provides valuable information for understanding the overall metabolite changes in rapeseed under N deficiency conditions, which may be beneficial for improving and producing new varieties of rapeseed capable of high yields under low N conditions.

## 1. Introduction

Nitrogen (N), a basic building block of fundamental biological molecules such as nucleic acids, amino acids, and proteins, and scores of secondary metabolites, plays crucial roles in ensuring high crop yields. Nitrogen deficiency can severely affect crop growth and development, such as by decreasing chlorophyll synthesis, promoting plant dwarfing, and decreasing yields [1,2,3]. In the major agricultural countries of the world, increasing the chemical N fertilization supply is an important way to ensure that crop production does not decrease. However, using too many chemical N fertilizers in the field has caused serious environmental problems, such as soil acidification, eutrophication, groundwater pollution, and global warming [3,4,5]. Therefore, it is necessary to understand how crops adapt to low N field conditions either physiologically or molecularly, and to find useful solutions for reducing the need for chemical N fertilization without decreasing crop production. 

Plants have developed precise regulatory mechanisms for soil N deficiency stress adaptation or tolerance, such as slowing growth, altering root architecture, enhancing N uptake and translocation, remobilizing N from old organs to young new tissues, improving the coordination of carbon (C) and N metabolism, and increasing flavonoid and derived compound accumulation [3,6,7]. Metabolites are important basic molecules that plants produce or consume during metabolic processes. When plants face stress conditions, they can increase or decrease the accumulation of stress-related metabolites in some key pathways to help adapt to stress conditions [8,9]. In the model plant species *Arabidopsis thaliana*, it has been proven in several studies that plants under N-deficient conditions can change the accumulation and redistribution of specific metabolites to integrate internal and external N availability signals, and then remodel root architecture [3,10,11]. C-terminally encoded peptides (CEPs) have emerged as regulators of systemic N signaling and are produced in roots experiencing N limitation in order to activate NO_3_^−^ transporters, such as NRT2.1, in roots, where NO_3_^−^ is plentiful [12,13,14]. Under low N conditions, the roots of plants can sense N deficiency and increase the biosynthesis of cytokinins, which acts as a root-to-shoot signal of high N availability and coordinates with the CEPs signal to regulate root development, especially the lateral root architecture, in order to adapt to a N-deficient environment [15]. Such root-shoot-root signaling is fundamental to ensure that plants make use of local N nutrients, but not only when there is a sufficient need for that N nutrient. The stress-related plant hormones jasmonic acid (JA) and salicylic acid (SA) also play important roles in affecting N uptake and allocation in plants. Wu et al. (2018) suggested that JA signaling could mediate large-scale systemic changes in N uptake and allocation in rice plants against stress conditions [16]. Plant flavonoid compounds also play essential roles in plants under N deficiency conditions. Flavonoids have been proven to play various crucial roles in plant growth, development, and defense [17,18]. For example, flavonoids can function as auxin transport inhibitors and act as positional signals that integrate hormonal and ROS pathways to affect the direction and rate of root development under many stress conditions [17,18]. It has been demonstrated that the lateral root primordia of *Arabidopsis* are rich in flavonoids, which suggests that flavonoids may play essential roles in plant root growth [19]. Anthocyanins belong to a specific branch of the flavonoid pathway [20,21]. Glucosinolates are a group of plant secondary metabolites in Brassica vegetables. Glucosinolates are N-rich metabolites and, therefore, sufficient N availability is essential for their biosynthesis. Glucosinolates can promote various kinds of stress resistance in plants, such as plant defense, drought stress, and salt stress [22,23]. Although many plant metabolites have been identified and characterized in many studies, the use of widely untargeted metabolomics methods to identify and analyze the changes in the metabolites in crop plants under N deficiency conditions is rare.

To cope with environmental N limitation, plants have evolved sophisticated mechanisms in unfavorable environments. These coordination mechanisms include regulating N uptake system activity and modulating root system architecture. Under N-deficient conditions, plant metabolites act as key molecules throughout various changes in physiology and molecular regulatory processes involved in plants under N deficiency. In recent years, advances in high-throughput metabolomic detection methods have led to large-scale screening and identification of metabolites in various plant issues [24]. In this work, we focus on metabolite content changes in the leaves and roots of rapeseed (*Brassica napus* L.) under N-deficient conditions. The objective of this research is to provide new insight into revealing the key metabolite pathways and metabolites of rapeseed leaves and roots, in order to cope with N deficiency stress. This information could be used for selecting new rapeseed varieties, and for managing rapeseed field nutrients.

## 2. Results 

### 2.1. Phenotypic Characterization of Rapeseed between the HN and LN Treatments

To investigate how the growth of rapeseed responds to N deficiency, we analyzed the phenotypic changes of rapeseed seedlings under HN and LN conditions. As shown in Figure 1A, the young leaves of the HN seedlings remained green, while the young leaves of the LN seedlings turned yellow. In addition, the LN seedlings accumulated anthocyanins in the stem, while none were observed in the HN seedlings. The roots of the LN seedlings turned slightly yellow, while the roots of the HN seedlings remained white (Figure 1A). The root biomass of the LN seedlings was significantly higher than that of the HN seedlings (Figure 1B). In contrast, the shoot biomass of the HN seedlings was higher than that of the LN seedlings (Figure 1C). The lengths of the total roots (Figure 1D) and primary roots (Figure 1E) of the LN seedlings were significantly greater than those of the HN seedlings. The lateral root number of the LN seedlings was greater than that of the HN seedlings (Figure 1F). Moreover, the root surface area and root volume of the LN seedlings were significantly greater than those of the HN seedlings (Figure 1G,H). Furthermore, the RD of the LN seedlings was shorter than that of the HN seedlings (Figure 1I). Taken together, these results indicated that rapeseed seedlings could alter their root architecture and synthesize stress-related components to adapt to LN conditions.

### 2.2. Multivariate Analysis of Identified Metabolites

Based on the results of metabolome detection, we carried out a principal component analysis (PCA) of the 574 identified metabolites. It was determined that PC1 (44.11%) clearly separated the leaf and root samples (Figure 2A), while PC2 (17.02%) also clearly distinguished between the HN and LN treatments (Figure 2A), and that the combined group was significantly different. These results revealed that the leaves and roots of rapeseed had distinct metabolite profiles. The PCA also indicated significant differences between the HN and LN treatments. Moreover, we also carried out a cluster analysis of the 574 detected metabolites. The results shown in Figure 2B suggested that the leaves and roots of rapeseed are two distinct groups in the metabolite clustering. The cluster analysis results also indicated that HN and LN treatments could significantly affect the metabolite clustering (Figure 2B). Together, PCA and cluster analysis suggested that the leaves and roots of rapeseed and the HN and LN treatments had distinct metabolite profiles.

### 2.3. Metabolic Changes in the Leaves and Roots of Rapeseed during the HN and LN Treatments

Based on the orthogonal projections to latent structures discriminant analysis (OPLS-DA) results, 175 metabolites differentially accumulated between HN and LN (|fold change| ≥ 2 and variable importance in projection (VIP ≥ 1) in leaf tissues, namely, 102 increased and 73 decreased metabolites under LN condition (Figure 3A). As shown in Figure 3B, 166 metabolites differentially accumulated between HN and LN (|fold change| ≥ 2 and VIP ≥ 1) in root tissues, namely, 77 increased and 89 decreased metabolites under LN condition. Among the differentially accumulated metabolites, the accumulation of 113 and 104 significantly changed in the leaves and roots, respectively, under the LN treatment, as compared to the HN treatment (Figure 3C). Moreover, the contents of 62 metabolites significantly changes in both the leaves and the roots during LN treatment (Figure 3C). These results indicated that metabolites differentially accumulated in the leaf and root tissues of rapeseed in response to LN treatment.

### 2.4. KEGG Analysis

The differentially accumulated metabolites in the leaves and roots were classified into six categories according to the KEGG classification results: organismal systems, metabolism, human diseases, genetic information processing, drug development and cellular processes (Figure 4A,B). According to the KEGG classification results, as shown in Figure 4, the number of secondary metabolic pathways was different between the leaves and roots during LN treatment. For example, there were 22 enriched secondary metabolic pathways in the leaves and 13 in the roots in the organismal systems classification. Furthermore, the statistics of KEGG enrichment analysis showed that the most significantly enriched pathways of differentially accumulated metabolites during LN treatment were biosynthesis of secondary metabolites and biosynthesis of phenylpropanoids (Figure 5A). In the leaf tissues, the differentially accumulated metabolites were mainly enriched in the flavonoid biosynthesis, biosynthesis of secondary metabolites, and biosynthesis of phenylpropanoids pathways (Figure 5A). Furthermore, in the root tissue, the differentially accumulated metabolites were largely enriched in the microbial metabolism in diverse environments, biosynthesis of amino acids, tryptophan metabolism, and lysine biosynthesis pathways (Figure 4B). These results suggested that the leaf tissues of *Brassica napus* L. primarily accumulated metabolites involved in the plant response to stress, while the metabolites in the root tissues were mainly involved in the response to plant development and rhizospheric microorganisms.

### 2.5. LN Treatment Increased Flavonoid and Sugar Accumulation in Rapeseed

Flavonoids have been demonstrated to protect plants from various biotic and abiotic stresses [12,17,25,26,27,28]. In the present study, our results showed that the flavonoid contents in the leaves under LN conditions were significantly different compared to those under HN conditions (Table 1); among them, 25 were increased (3 anthocyanins, 22 flavonols), and 9 were decreased (1 anthocyanin, 3 flavones, 5 flavanones). In the root tissues, under LN conditions, there were 25 flavonoids whose content significantly changed compared to those under the HN conditions (Table 1). Namely, 13 increased (2 anthocyanins, 6 flavonols, 1 flavone C-glycoside, and 4 flavanones) and 12 decreased (3 anthocyanins, 5 flavones, and 4 flavone C-glycosides). The above results suggested that more flavonoids differentially accumulated in the leaves than in the roots under LN conditions.

In addition to flavonoids, sugars have also been proven to play crucial roles in plants in response to various stress conditions [7,29,30,31]. In this work, except for 2-deoxyribose 1-phosphate, whose content decreased in the leaves under LN conditions, the other detected sugars all increased in the leaves or roots under LN conditions compared to HN conditions (Table 2). Six sugars significantly increased in the leaves under LN conditions, while four sugars increased in the roots. Among these sugars, D-sucrose, D-glucuronic acid, and DL-arabinose all increased both in the leaves and roots in response to LN conditions compared to HN, which suggested that those sugars could be stress-related sugars in rapeseed. Trehalose 6-phosphate, which is involved in source transport and feedback regulation in leaves, might be a signaling molecule in plants [32]. In the present work, trehalose 6-phosphate significantly increased in the leaves compared to the roots, but only under LN conditions (Table 2), which indicated that LN treatment could significantly affect photosynthesis and regulate trehalose 6-phosphate synthesis.

### 2.6. LN Treatment Affects Amino Acids and Their Derivative Metabolites in Rapeseed

Nitrogen is the basic elemental ingredient from which amino acids are synthesized. In this work, 12 amino acids (4), and their derivatives (8), increased significantly in the leaves under LN conditions, while 9 amino acids and their derivatives decreased compared to those under the HN conditions (Table 3). In the roots, 12 amino acid-related metabolites increased, and 14 decreased (Table 3). Among these different amino acid-related metabolites, six metabolites increased, and four metabolites decreased (asterisks in Table 3) in the leaves and roots, respectively. Moreover, the aspartic acid content increased 42-fold in the leaves and decreased 3-fold in the roots under LN conditions. Furthermore, the L-Cysteine content increase 104-fold in leaves in HN conditions compared to the LN conditions. The L-alanine content increased 3-fold in the roots and decreased 6-fold in the leaves under LN conditions (Table 3). The above results suggested that more amino acids and their derivative metabolites were differentially accumulated in the leaves compared to the roots under LN conditions.

### 2.7. LN Treatment Alter Phytohormone Accumulation in Rapeseed

Plant hormones play crucial roles in plant resistant to environment stresses. In the present study, we detected five significantly different accumulated phytohormones, including indole carboxylic acid (ICA), Methyl indole-3-acetate (MeJA), indole 3-acetic acid (IAA), gibberellin A20 (GA20) and N-[(−)-jasmonoyl]-(L)-isoleucine (JA-L-Ile) (Table 4). Along these phytohormones, ICA and MeJA were up-regulated in HN leaves compared to LN leaves, while IAA, gibberellin A20 (GA20) and JA-L-Ile were down-regulated in HN leaves. In roots, there were only three phytohormones identified, including ICA, trans-zeatin 9-O-glucoside (zeatin) and salicylic acid O-glucoside (SA). All three phytohormones were down-regulated in HN roots compared to LN roots (Table 4). These results indicated that LN stress could promote rapeseed accumulation of different kinds of plant hormones in the leaf and root to regulate self-growth, and to increase resistance ability to adaptation LN stress.

### 2.8. LN Treatment Can Modulate the Carbon-Nitrogen Metabolite Balance in the Leaves and Roots of Rapeseed and Increase the Accumulation of Multiple Stress-Related Metabolites

Carbon and N are the most important fundamental elements of organic compounds in plants. Shoots and roots depend on each other by exchanging C and N through complex physiological transport systems, such as the xylem and phloem. Plants have evolved complex and sophisticated mechanisms to regulate C and N metabolism to counter various environmental stresses and ensure plant growth and development. In this work, when rapeseed seedlings were grown under N-deficient conditions, the sugar metabolites and tricarboxylic acid cycle metabolite accumulation in the leaves and roots were reduced compared to those in the seedlings grown under a normal N supply (Figure 6). In contrast to the C metabolites, most N metabolites were decreased in the leaves and roots of seedlings under the normal N supply compared to the low N supply (Figure 6). Interestingly, as important stress-related metabolites, flavonoids increased in the roots of seedlings compared to the leaves of seedlings under both normal and low N supplies (Figure 6). Furthermore, phenylalanine showed the greatest decrease in content in the low N-supplied leaves of seedlings, and an increase in the roots of low N-supplied seedlings compared to the normal N-supplied seedlings (Figure 6). These results indicated that rapeseed seedlings could modulate their C-N metabolite balance and reassign the C-N and stress-related metabolites in the leaves and roots in order to adjust to N-deficient conditions.

## 3. Discussion

### 3.1. N Deficiency Led to Architectural Changes in Rapeseed Roots

As fixed organisms, plants have evolved sophisticated mechanisms to cope with various environmental stresses. In this work, we analyzed the phenotypic changes of rapeseed under HN and LN conditions. The results showed that the shoot and root biomass were opposite in the HN and LN seedlings (Figure 1B,C). These results suggested that rapeseed might inhibit shoot growth and increase root numbers and root density to reduce nutrient assimilation and increase root nutrient absorption, in order to adapt to N deficiency. Our previous studies have proven that the elongation zone cells of the roots of seedlings under the LN treatment of rapeseed were significantly elongated compared to the cells of the roots of seedlings under the HN treatment [3]. In the present work, we found that the content of gibberellic acid (GA) in the roots of seedlings under the LN treatment was five-fold higher than that in the roots of seedlings under the HN treatment (Table 4), which might partly explain the cell length changes. Moreover, our previous research also demonstrated that the meristematic zone of root tips increased by 27.6% in roots of seedlings under the LN treatment [3]. These results indicated that cell expansion, cell division, and increased GA content might contribute to the increased root length of rapeseed under LN conditions. Future research could focus on discovering the key transcription factors that regulate the rapeseed response to N deficiency.

### 3.2. N Deficiency Led to Substantial Changes in Metabolite Contents in the Leaves and Roots of Rapeseed

Nitrogen is the essential constitutive nutrition element that drives and promotes plant growth and crop yields. As a typical winter crop species, rapeseed needs a large amount of N fertilizer to maintain normal growth and development during the whole rapeseed growth stage, especially before the bolting stage. If rapeseed seedlings cannot assimilate enough N fertilizer, they cannot undergo proper vegetative growth and remain safe from winter frost, which ultimately affects crop yields. However, the N nutrient supply in the soil is highly dynamic and can cause a large number of nitrate zones to accumulate in the soil, while long-term use of N fertilizer can also pollute the air, groundwater, and soil. To respond to N deficiency, rapeseed usually delays its growth or accumulates stress-related metabolites such as abscisic acid (ABA), flavonoids, and JA to maintain basic growth and resistance to oxidation damage [16,18,20,23]. Prior to this study, several model and crop plant species were extensively studied for their changes in metabolites in response to various nutrition deficiency conditions [33,34,35]. However, the oil crop species rapeseed has not been thoroughly investigated to determine how its metabolite levels respond to N deficiency. Here, we used a high-throughput metabolomic detection method to elucidate the metabolite differences in the leaves and roots of rapeseed seedlings between normal N and low N conditions. Our results revealed 574 differentially accumulated metabolites in the leaves and roots, and PCA of these 574 detected metabolites indicated that the metabolites in the leaves and roots belonged to two distinct metabolite cluster groups under HN and LN conditions. These findings suggested that the leaves and roots of rapeseed could accumulate metabolites belonging to specific groups to protect against N deficiency stress.

### 3.3. Flavonoids Play an Important Protective Role in Rapeseed under N Starvation Conditions

The main skeleton compounds of the phenylalanine and flavonoid biosynthesis pathway are widely conserved throughout the plant kingdom. It is known that petunidin, malvidin, and delphinidin exhibit purple and dark colors, whereas cyanidin and pelargonidin exhibit red colors. In our study, petunidin and delphinidin were 17.3-and 7.6-fold higher in the leaves of seedlings under the HN treatment than in the leaves of seedlings under the LN treatment (Table 1), respectively. These results may partly explain why rapeseed did not turn red or purple under N deficiency stress. In addition to these common color-related anthocyanin metabolites, using a widely targeted metabolomic approach, we investigated and detected changes in the contents of other known metabolites of the flavonoid pathway (Figure 6). Our results showed that the contents of most flavonoid metabolites in the leaves of seedlings under the HN treatment were higher than those in the leaves of seedlings under the LN treatment. Conversely, the contents of most flavonoid metabolites in the roots of seedlings under the HN treatment were lower in abundance than those in the roots of seedlings under the LN treatment were (Figure 6). These results suggested that flavonoids play a crucial role in the roots of rapeseed seedlings against N deficiency. One of the explanations for the increased accumulation of flavonoids in the roots of seedlings under the LN treatment of rapeseed, when compared to the roots of seedlings under the HN treatment, is that flavonoids might mitigate reactive oxygen species (ROS) accumulation [18,21,31]. Therefore, we propose that increasing the biosynthesis of flavonoids could be a potential approach to protect root cells from oxidative damage under N deficiency. Moreover, the rutin content in the roots was also higher than that in the leaves in the rapeseed seedlings after HN or LN treatments (Table 1), which indicated that rutin might also play an important role in the antioxidant reaction in the rapeseed seedling roots in response to N deficiency. However, previous studies suggested that rutin usually has a high content in the flowers, leaves, pericarps, seeds, and fruits, but not usually in the roots [36,37]. Further studies could focus on elucidating the molecular regulatory mechanism underlying why the rutin content in rapeseed roots is higher than that in leaves under normal or N-deficient conditions.

### 3.4. Amino Acids and Their Derivative Metabolites May Enhance Rapeseed Resistance to N Starvation Conditions

It has been proven that L-alanine-rich proteins can regulate metal ion homeostasis, oxidative stress, and hypoxia stress [38,39]. In this work, the L-alanine content increased five-fold in leaves and three-fold in roots under LN conditions (Table 3), which suggested that L-alanine mainly participated in the response to LN stress in the roots of rapeseed. Aspartic acid has also been suggested to be involved in helping plants protect against diverse stress conditions, such as salt stress, heavy metal stress, osmotic stress, and cold stress [40,41,42]. In the present study, the aspartic acid content decreased significantly in the leaves, but increased in the roots under LN conditions (Table 3). Our results suggested that aspartic acid may play an important role in roots in response to LN stress. Further research could focus on revealing the molecular regulatory network between the N deficiency signal and stress-related amino acid biosynthesis pathways.

### 3.5. Plant Hormones Function Synergistically to Regulate and Maintain Rapeseed Growth under N Starvation

Previous studies have shown that plant hormones play crucial roles in plants which are faced with environmental stress [18,43]. In the present work, we detected seven kinds of hormone metabolites that significantly changed in the leaves and roots of rapeseed seedlings under normal and N-deficient conditions. Among these detected hormone metabolites, indole-carboxylic acid (ICA) and methyl indole-3-acetate showed a 300-and 5.8-fold increases in accumulation in the leaves of seedlings under the HN treatment compared to the leaves of seedlings under the LN treatment, respectively (Table 4). It has been reported that ICA is the basal component involved in the plant cell wall against bacterial infection [44]. This result may provide a new clue as to why the LN rapeseed seedlings are more susceptible to infection by plant pathogens than the HN seedlings were. Moreover, the contents of indole 3-acetic acid (IAA), gibberellin (GA_20_), and jasmonoyl-(L)-isoleucine (JA-L-IIe) in the leaves of seedlings under the HN treatment were 10-fold different from those in the leaves of seedlings under the HN treatment (Table 4). A previous study demonstrated that JA is essential for both resistance to leaf disease and leaf senescence [16,45]. Additionally, IAA and GA have been proven to delay leaf senescence [46]. It seems contradictory that IAA, GA, and JA accumulate in the leaves under N deficiency simultaneously. One possible explanation is that N deficiency may induce and promote the biosynthesis and location of these key hormones in the leaf to coordinate and maintain leaf growth, and to increase resistance to environmental stress damage. Further research could focus on determining the signal transduction pathway crosstalk between N biosynthesis and metabolism and these key hormones to elucidate how N deficiency affects the synthesis and distribution of these hormones. In this work, ICA, trans-zeatin 9-O-glucoside (ZT), and SA O-glucoside were significantly decreased in the roots of seedlings under the LN treatment compared to the roots of seedlings under the HN treatment (Table 4). Among the three hormones, ICA was detected in both the leaves and the roots, but the accumulation patterns were opposite. These results suggested that rapeseed might synthesize and transport more ICA to the roots to protect them from infection by soil pathogens and to better take up nutrients to maintain growth under N deficiency. The increased content of the plant pathogen-related hormones ZT and SA in the roots of seedlings under the LN treatment also supported our views that rapeseed root cells could be protected from damage caused by soil pathogens via accumulating ICA, ZT, and SA, which have been proven to play important roles in resistance to plant pathogens [16,44]. Further research could focus on elaborating the molecular regulatory mechanism between N deficiency and plant pathogen-related hormones.

## 4. Conclusions

In the present study, the changes in metabolites categories and abundance in leaf and root of rapeseed seedlings after LN treatment were comprehensively analyzed, respectively. The results showed that the significantly deferentially accumulated metabolites were involved in four primary metabolic pathways, namely sucrose, phenylalanine, amino acid, and tricarboxylic acid cycle metabolism. Notably, we found that plant hormones have distinct accumulation patterns in rapeseed, and that these coordinate to play crucial roles in both maintaining growth and protecting against damage from plant disease under HN and LN conditions. Moreover, our results indicated that flavonoid compounds, especially anthocyanins and rutin, may play important roles in increasing root cell resistance to oxidative damage and soil pathogen infection. Our results provide worthwhile information for understanding how rapeseed plants adapt to N deficiency conditions by accumulating different kinds of metabolites. Our results also provide a scientific theory foundation for improving and producing new varieties of rapeseed capable of high yields under N deficiency conditions.

## 5. Materials and Methods

### 5.1. Rapeseed Seedlings and N-Deficient Treatment

We used the *Brassica napus* L. cultivar of ZS11 as the experimental material and subjected the rapeseed seedlings to a hydroponic nutrient solution with normal N and N-deficient solutions. After sterilization with 5% NaClO solution, the seeds of ZS11 were sown on a growth plate with 1/4-strength modified Hoagland nutrient solution for predetermination. After 5 days, we selected uniformly growing seedlings, divided them into 2 groups, and placed them in a plant culture chamber containing 1/2-strength modified Hoagland nutrient solution (pH 5.8, normal N supply) and 1/2-strength modified Hoagland nutrient solution (pH 5.8, 1/50 of normal N supply) under a 16/8 h light/dark photoperiod at 28 °C for 14 d. After 14 d of treatment, the leaves and roots of the seedlings in the 2 treatment groups were collected immediately, frozen in liquid nitrogen, and stored at −80 °C until further analysis. Each analysis was performed in biological triplicate.

### 5.2. Phenotypic Characterization

To assess the phenotypic and physiological changes of ZS11 at the seedling stage in response to N-deficient conditions, we used the same treatment method as described above to treat the ZS11 seedlings. The images of rapeseed seedlings from normal and low N treatments were recorded with a camera (Nikon D800, Tokyo, Japan). The rapeseed roots and shoots were separately harvested and dried in an oven at 75 °C to determine biomass. Roots from each of the treatments were spread out on a transparent plastic tray with water and imaged with a scanner (Epson 700V J221A, Nagano, Japan), after which the total root length (RL), total root surface area (RS), total root volume (RV), and root average diameter (RD) were analyzed using WinRHIZO software (Pro, 2012b, Regent Instruments Inc, Quebec City, QC, Canada), as described before [47]. In addition, the primary root length was measured with a ruler. The differences were statistically assessed using a Student’s *t*-test (** *p* < 0.01, * *p* < 0.05).

### 5.3. Plant Sample Preparation and Extraction of Secondary Metabolites

The freeze-dried leaves and roots were crushed using a mixer mill (MM 400, Retsch) with zirconia beads for 1.5 min at 30 Hz. Then, 100 mg powder was weighed and extracted overnight at 4 °C with 1.0 mL of 70% aqueous methanol. Following centrifugation at 10,000× *g* for 10 min, the extracts were absorbed (CNWBOND Carbon-GCB SPE Cartridge, 250 mg, 3 mL; ANPEL, Shanghai, China, www.anpel.com.cn/cnw, accessed on 17 April 2022) and filtered (SCAA-104, 0.22 μm pore size; ANPEL, Shanghai, China, http://www.anpel.com.cn/, accessed on 17 April 2022), before liquid chromatographic mass spectrometry (LC–MS) analysis.

The sample extracts were analyzed using an LC–ESI–MS/MS system (HPLC, Shim-pack UFLC Shimadzu CBM30A system, www.shimadzu.com.cn/, accessed on 17 April 2022; MS, Applied Biosystems 6500 Q TRAP, www.appliedbiosystems.com.cn/, accessed on 17 April 2022). The HPLC analytical conditions were as follows: column, Waters ACQUITY UPLC HSS T3 C18 (1.8 µm, 2.1 mm * 100 mm); solvent system, water (0.04% acetic acid):acetonitrile (0.04% acetic acid); gradient program, 95:5 *v*/*v* at 0 min, 5:95 *v*/*v* at 11.0 min, 5:95 *v*/*v* at 12.0 min, 95:5 *v*/*v* at 12.1 min, and 95:5 *v*/*v* at 15.0 min; flow rate, 0.40 mL/min; temperature, 40 °C; and injection volume: 2 μL. The effluent was alternatively analyzed through an ESI-triple quadrupole-linear ion trap (Q TRAP)-MS.

### 5.4. LC–MS Analysis

Subsequently, LIT and triple quadrupole (QQQ) scans were acquired on a triple quadrupole-linear ion trap mass spectrometer (Q TRAP), API 6500 Q TRAP LC/MS/MS System, equipped with an ESI Turbo Ion-Spray interface, operating in positive ion mode and controlled by Analyst 1.6 software (AB SCIEX). The ESI source operation parameters were as follows: ion source, turbo spray; source temperature 500 °C; ion spray voltage 5500 V; ion source gas I, gas II, and curtain gas, 55, 60, and 25.0 psi, respectively; and collision gas, high. Instrument tuning and mass calibration were performed with 10 and 100 μmol/L polypropylene glycol solutions in QQQ and LIT modes, respectively. The QQQ scans were acquired as MRM experiments with collision gas set to 5 psi. The DP and CE for individual MRM transitions were performed with further DP and CE optimization. A specific set of MRM transitions was monitored for each period according to the metabolites eluted within the period.

### 5.5. Metabolite Profiling

Metabolite profiling was carried out using a widely targeted metabolome method by Wuhan Metware Biotechnology Co., Ltd. (Wuhan, China) (http://www.metware.cn/, accessed on 17 April 2022). The freeze-dried samples were extracted as previously described, before analysis using an LC–electrospray ionization (ESI)-MS/MS system. The extracts were absorbed (CNWBOND Carbon-GCB SPE Cartridge, 250 mg, 3 mL; Shanghai ANPEL Scientific Instrument Co., Ltd., Shanghai, China). Metabolites were quantified using a multiple reaction monitoring method. PCA was used to analyze the variability between groups and within groups. Partial least squares-discriminant analysis (PLS-DA) was performed to differentially accumulated metabolites (DAMs). The HCA (hierarchical cluster analysis) results of samples and metabolites were presented as heatmaps with dendrograms, while pearson correlation coefficients (PCC) between samples were calculated by using the cor function in R version 4.1.0), and then presented as heatmaps. Both HCA and PCC were carried out using the R package ComplexHeatmap. For HCA, normalized signal intensities of metabolites (unit variance scaling) are visualized as a color spectrum. Significantly regulated metabolites between groups were determined by VIP (VIP ≥ 1) and absolute Log2FC (|Log2FC| ≥ 1.0). The VIP values were extracted from OPLS-DA results, which also contain score plots and permutation plots, generated using R package MetaboAnalystR. The data was log transformed (log2) and underwent mean centering before OPLS-DA. In order to avoid overfitting, a permutation test (200 permutations) was performed. Identified metabolites were annotated using the KEGG Compound database (http://www.kegg.jp/kegg/compound, accessed on 17 April 2022), and annotated metabolites were then mapped to the KEGG Pathway database (http://www.kegg.jp/kegg/pathway.html, accessed on 17 April 2022). Pathways with significantly regulated metabolites mapped to them were then fed into MSEA (metabolite sets enrichment analysis), and their significance was determined by a hypergeometric test’s *p*-values. 

## Figures and Tables

**Figure 1 ijms-23-05784-f001:**
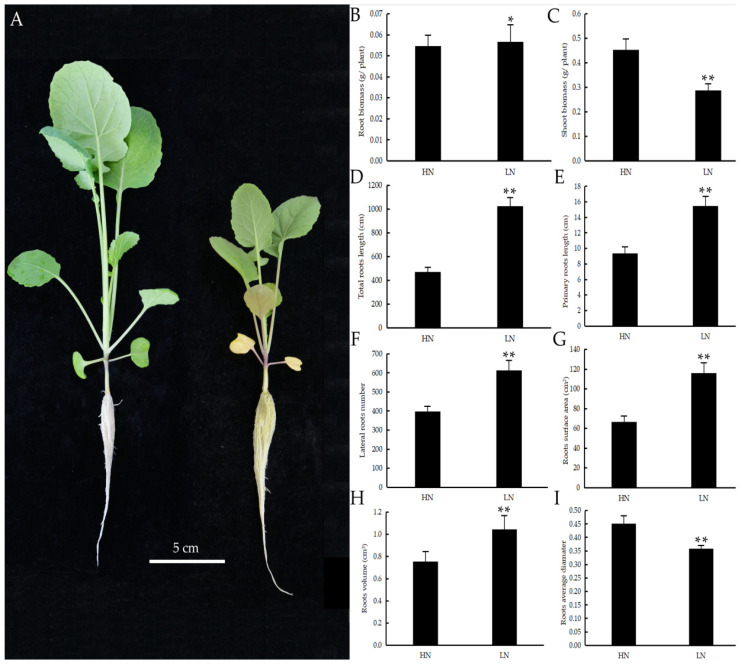
Phenotypic characterizations of rapeseed under HN and LN treatment. (**A**) Phenotypes of rapeseed seedlings under HN and LN treatment; root biomass (**B**), shoot biomass (**C**), RL (**D**), primary root length (**E**), lateral root number (**F**), root surface area (**G**), root volume (**H**), and RD (**I**) of rapeseed seedlings under HN and LN treatment. Each point represents the mean value of three independent experiments performed in triplicate + the SE. Statistically significant differences were assessed using Student’s *t*-test (* *p* < 0.01; ** *p* < 0.005).

**Figure 2 ijms-23-05784-f002:**
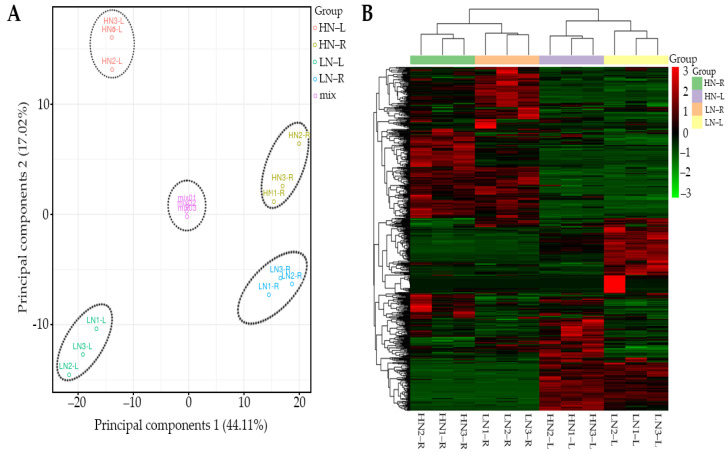
PCA and heatmap analysis of all metabolites detected in the leaves and roots of rapeseed under HN or LN conditions. (**A**) PCA score plots were derived from the relative contents of all detected metabolites by LC–ESI–MS/MS, with six replicates per treatment. (**B**) Heatmap showing 574 differentially accumulated metabolites of the leaves and roots between HN and LN conditions. The values of the metabolites were normalized and are shown as a color scale. The high and low metabolite levels are represented as red and green scales, respectively.

**Figure 3 ijms-23-05784-f003:**
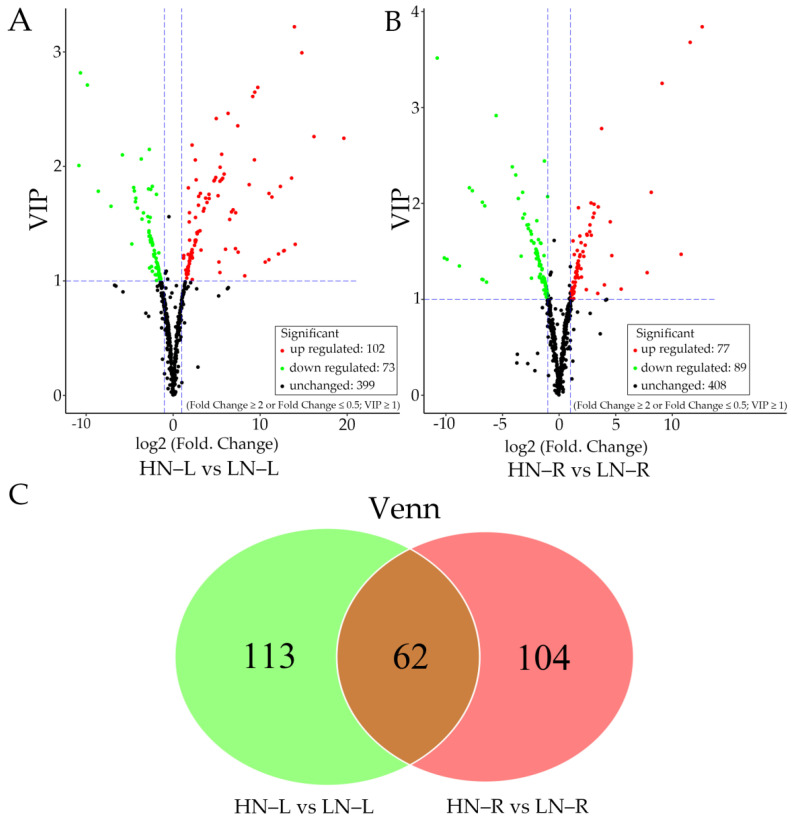
Volcano plot and Venn diagram analysis of differentially accumulated metabolites of rapeseed. (**A**) Volcano plot of differentially accumulated metabolites of rapeseed between HN-L and LN–L; (**B**) Volcano plot of differentially accumulated metabolites of rapeseed between HN-R and LN–R. The red points, green points, and black points indicate metabolites whose accumulation significantly increased, decreased, and was not significantly different, respectively. (**C**) Venn diagrams of the numbers of significantly differentially accumulated metabolites in leaves and roots between HN and LN conditions. The green and pink represent significantly differentially accumulated metabolites only in HN/LN–L and HN/LN–R, respectively. The brown represents significantly differentially accumulated metabolites detected both in HN/LN–L and HN/LN-R.

**Figure 4 ijms-23-05784-f004:**
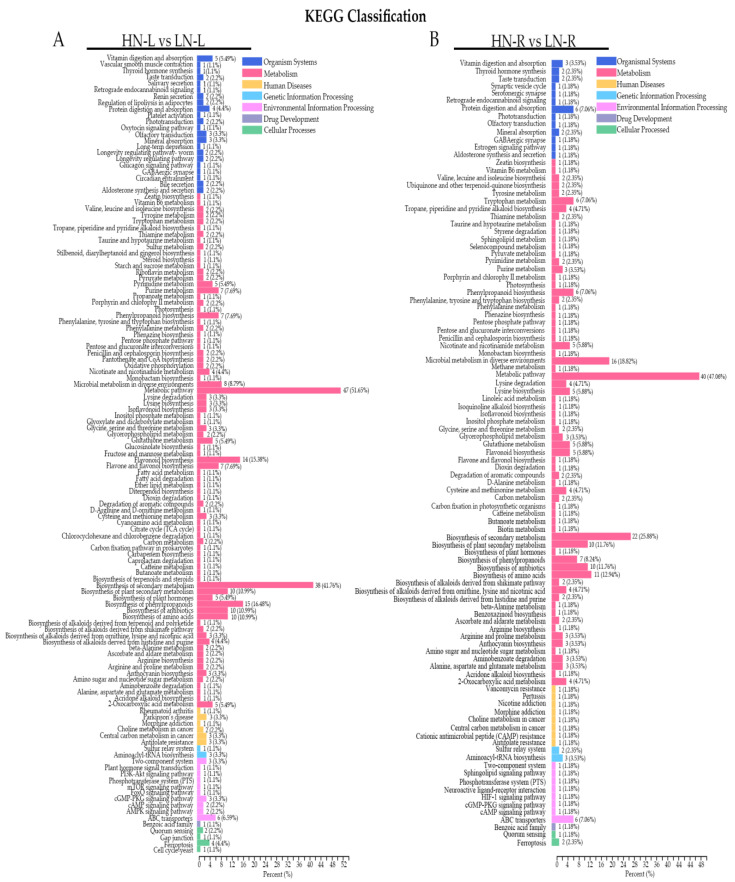
KEGG analysis of the detected metabolites. Functional annotations of the detected metabolites based on metabolite KEGG categorization of the HN-L vs. LN-L (**A**) and HN-R vs. LN-R (**B**).

**Figure 5 ijms-23-05784-f005:**
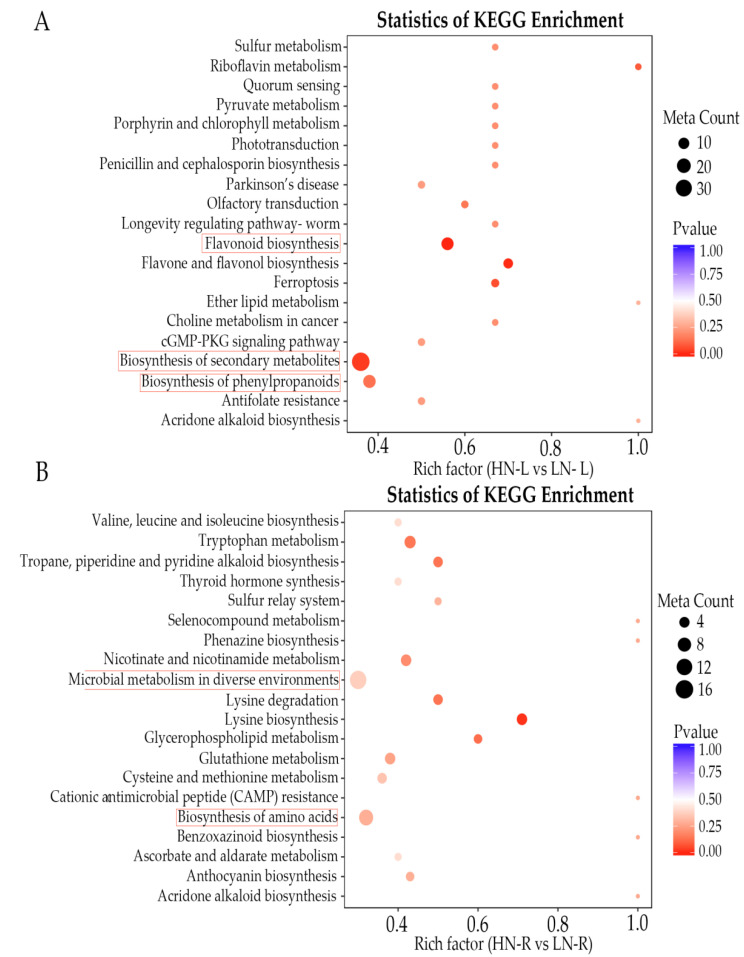
Enrichment analysis of the detected metabolites. Significantly enriched KEGG pathways (*p* < 0.05) from detected metabolites in HN-L vs. LN-L (**A**) and HN-R vs. LN-R (**B**). The x-axis shows the fold enrichment. The y-axis shows the metabolic pathway terms. The size of the plotted circle indicates the number of metabolites. The red solid box represents the key enrichment pathway.

**Figure 6 ijms-23-05784-f006:**
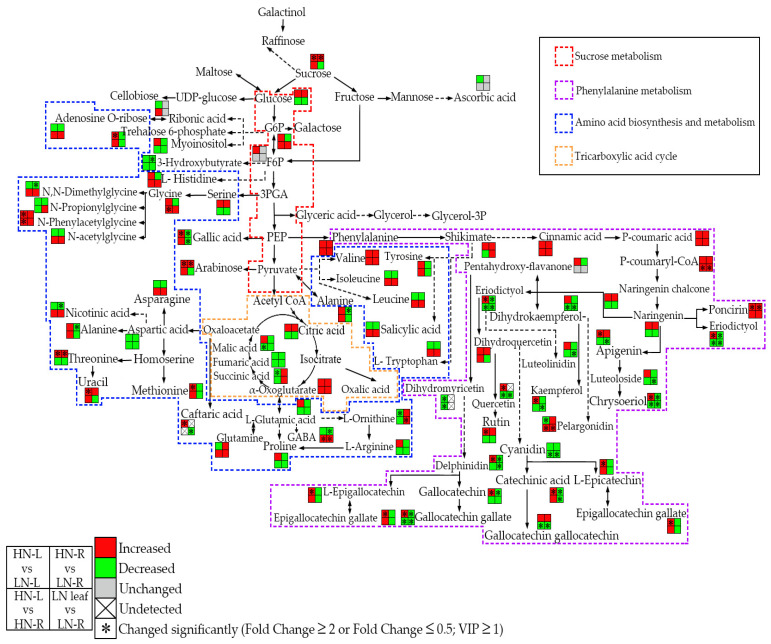
Schematic presentation of the pathway of certain important metabolites whose accumulation is affected by HN or LN conditions in the leaves and roots of rapeseed. The red represents an increase, while the green represents a decrease. The colored dotted boxes represent important metabolic pathways. The solid black arrow represents a direct reaction, while the black dotted arrow represents an indirect reaction. The asterisk marks indicate that the relative levels of metabolites were significantly different (fold change > 2 or fold change < 5). VIP > 1.

**Table 1 ijms-23-05784-t001:** Fold change of flavone and flavonoid metabolites detected in the HN leaf when compared to the LN leaf and, HN root compared to the LN root, respectively. Asterisk marks indicate that the metabolite was detected both in the leaf and root.

**HN-L vs. LN-L**				
**ID**	**Class**	**Fold Change**	**Compounds**	**Type**
Bra0152	Anthocyanins	3.728	Delphinidin 3-O-glucoside (Mirtillin)	up
* Bra0166	Anthocyanins	17.390	Petunidin 3-O-glucoside	up
* Bra0194	Anthocyanins	7.650	Delphinidin	up
Bra0232	Flavone	29.870	Selgin 5-O-hexoside	up
* Bra0256	Flavone	177.398	Apigenin 7-rutinoside (Isorhoifolin)	up
* Bra0282	Flavone	4.529	Chrysoeriol 7-O-hexoside	up
Bra0284	Flavone	7.519	Selgin O-malonylhexoside	up
Bra0593	Flavone	6.265	Acacetin O-acetyl hexoside	up
Bra0636	Flavone	3.709	Velutin O-glucuronic acid	up
Bra0641	Flavone	47.791	Chrysoeriol 5-O-hexoside	up
Bra0648	Flavone	649.644	Chrysoeriol 7-O-rutinoside	up
Bra0692	Flavone	3.292	Butin	up
Bra0230	Flavonol	41.507	Quercetin 3-O-rutinoside (Rutin)	up
* Bra0245	Flavonol	143.514	Kaempferol 3-O-robinobioside (Biorobin)	up
Bra0249	Flavonol	38.440	Quercetin 7-O-rutinoside	up
* Bra0255	Flavonol	145.810	Kaempferol 3-O-rutinoside (Nicotiflorin)	up
Bra0258	Flavonol	172.283	methylQuercetin O-hexoside	up
* Bra0261	Flavonol	27.174	Isorhamnetin 5-O-hexoside	up
Bra0286	Flavonol	42.274	Isorhamnetin O-hexoside	up
Bra0314	Flavonol	13.310	Kaempferol	up
Bra0629	Flavonol	51.798	Quercetin 3-O-glucoside (Isotrifoliin)	up
Bra0630	Flavonol	428.117	Quercetin 7-O-β-D-Glucuronide	up
* Bra0658	Flavonol	58.168	Quercetin 4′-O-glucoside (Spiraeoside)	up
Bra0680	Flavonol	35.875	Quercetin	up
Bra0682	Flavonol	79.744	Morin	up
* Bra0178	Anthocyanins	0.146	Pelargonidin 3-O-beta-D-glucoside (Callistephin chloride)	down
Bra0277	Flavone	0.317	Apigenin 7-O-glucoside (Cosmosiin)	down
Bra0320	Flavonol	0.141	Isorhamnetin	down
Bra0614	Flavonol	0.001	Dihydromyricetin	down
Bra0370	Flavanone	0.320	Xanthohumol	down
Bra0611	Flavanone	0.315	Afzelechin (3,5,7,4′-Tetrahydroxyflavan)	down
Bra0678	Flavanone	0.191	Liquiritigenin	down
Bra0342	Isoflavone	0.007	Biochanin A	down
Bra0364	Isoflavone	0.037	Rotenone	down
**HN-R vs. LN-R**				
**ID**	**Class**	**Fold Change**	**Compounds**	**Type**
Bra0554	Anthocyanins	6.529	Cyanidin O-syringic acid	up
Bra0151	Anthocyanins	2.347	Cyanidin 3,5-O-diglucoside (Cyanin)	up
* Bra0265	Flavone	1759.926	Tricin 5-O-rutinoside	up
* Bra0245	Flavonol	282.185	Kaempferol 3-O-robinobioside (Biorobin)	up
* Bra0255	Flavonol	219.556	Kaempferol 3-O-rutinoside (Nicotiflorin)	up
Bra0626	Flavonol	6.621	Fustin	up
* Bra0239	Flavonol	3.191	Isorhamnetin 3-O-neohesperidoside	up
* Bra0658	Flavonol	2.211	Quercetin 4′-O-glucoside (Spiraeoside)	up
Bra0283	Flavone C-glycosides	2.323	Apigenin 8-C-pentoside	up
Bra0652	Flavanone	25.210	Hesperetin 7-rutinoside (Hesperidin)	up
Bra0663	Flavanone	16.040	Hesperetin 7-O-neohesperidoside (Neohesperidin)	up
* Bra0677	Flavanone	10.653	Isosakuranetin-7-neohesperidoside (Poncirin)	up
Bra0647	Flavanone	3.206	Hesperetin O-malonylhexoside	up
* Bra0194	Anthocyanins	0.0105	Delphinidin	down
* Bra0178	Anthocyanins	0.213	Pelargonidin 3-O-beta-D-glucoside (Callistephin chloride)	down
* Bra0166	Anthocyanins	0.282	Petunidin 3-O-glucoside	down
Bra0301	Flavone	0.0005	Tricetin O-malonylhexoside	down
* Bra0282	Flavone	0.0117	Chrysoeriol 7-O-hexoside	down
Bra0664	Flavone	0.184	Tricin 7-O-hexoside	down
Bra0293	Flavone	0.237	Apigenin O-malonylhexoside	down
* Bra0261	Flavonol	0.009	Isorhamnetin 5-O-hexoside	down
Bra0182	Flavone C-glycosides	0.009	Eriodictiol 6-C-hexoside 8-C-hexoside-O-hexoside	down
Bra0197	Flavone C-glycosides	0.009	Eriodictiol C-hexosyl-O-hexoside	down
* Bra0229	Flavone C-glycosides	0.255	Vitexin 2″-O-beta-L-rhamnoside	down
Bra0612	Flavone C-glycosides	0.329	Luteolin C-hexoside	down

**Table 2 ijms-23-05784-t002:** Fold change of sugars metabolites detected in the HN leaf when compared to the LN leaf and, HN root compared to the LN root, respectively. Asterisk marks indicate that the metabolite was detected both in the leaf and root.

**HN-L vs. LN-L**				
**ID**	**Class**	**Fold Change**	**Compounds**	**Type**
* Bra0436	Carbohydrates	25.867	D-(+)-Sucrose	up
Bra0437	Carbohydrates	31.017	L-Gulonic-γ-lactone	up
* Bra0443	Carbohydrates	8.515	D-glucoronic acid	up
* Bra0445	Carbohydrates	5.894	DL-Arabinose	up
Bra0449	Carbohydrates	3.364	Trehalose 6-phosphate	up
Bra0453	Carbohydrates	3.854	L-Fucose	up
Bra0456	Carbohydrates	0.160	2-Deoxyribose 1-phosphate	down
**HN-R vs. LN-R**				
**ID**	**Class**	**Fold Change**	**Compounds**	**Type**
* Bra0443	Carbohydrates	11.008	D-glucoronic acid	up
* Bra0436	Carbohydrates	2.702	D-(+)-Sucrose	up
* Bra0445	Carbohydrates	2.286	DL-Arabinose	up
Bra0447	Carbohydrates	2.212	D-(+)-Glucono-1,5-lactone	up

**Table 3 ijms-23-05784-t003:** Fold change of amino acid metabolites detected in the HN leaf when compared to the LN leaf, and HN root compared to the LN root, respectively. Asterisk marks indicate that the metabolite was detected both in the leaf and root.

**HN-L vs. LN-L**				
**ID**	**Class**	**Fold Change**	**Compounds**	**Type**
* Bra0056	Amino acids	2.922	2-Aminoadipic acid (L-Homoglutamic acid)	up
Bra0082	Amino acids	3.617	L-Methionine	up
* Bra0421	Amino acids	4.185	L-Threonine	up
Bra0435	Amino acids	104.407	L-Cysteine	up
* Bra0057	Amino acids	41.650	Aspartic acid	up
Bra0222	Amino acids derivative	3.361	N-Phenylacetylglycine	up
Bra0440	Amino acids derivative	17.621	L-Saccharopine	up
* Bra0451	Amino acids derivative	5.508	Allysine(6-Oxo DL-Norleucine)	up
Bra0483	Amino acids derivative	3.663	γ-Glu-Cys	up
* Bra0487	Amino acids derivative	90.556	(−)-3-(3,4-Dihydroxyphenyl)-2-methylalanine	up
* Bra0515	Amino acids derivative	6.176	3-Hydroxy-3-methylpentane-1,5-dioic acid	up
* Bra0656	Amino acids derivative	3.674	N-(3-Indolylacetyl)-L-alanine	up
Bra0009	Amino acids	0.018	L(+)-Ornithine	down
* Bra0431	Amino acids	0.001	L-(−)-Cystine	down
Bra0101	Amino acids derivative	0.196	N-Acetyl-L-glutamic acid	down
Bra0439	Amino acids derivative	0.300	L-Glutamine O-hexside	down
* Bra0470	Amino acids derivative	0.179	Glutathione oxidized	down
* Bra0510	Amino acids derivative	0.146	L-alanine	down
* Bra0553	Amino acids derivative	0.263	Asp-phe	down
* Bra0560	Amino acids derivative	0.079	N-Acetylmethionine	down
Bra0561	Amino acids derivative	0.372	2,3-dimethylsuccinic acid	down
**HN-R vs. LN-R**				
**ID**	**Class**	**Fold Change**	**Compounds**	**Type**
Bra0421	Amino acids	5.571	L-Threonine	up
Bra0043	Amino acids	3.255	L-Homocitrulline	up
Bra0426	Amino acids	3.105	L-Citrulline	up
* Bra0056	Amino acids	2.208	2-Aminoadipic acid (L-Homoglutamic acid)	up
Bra0503	Amino acid derivatives	13.495	S-(5’-Adenosy)-L-homocysteine	up
* Bra0487	Amino acid derivatives	8.523	(−)-3-(3,4-Dihydroxyphenyl)-2-methylalanine	up
Bra0050	Amino acid derivatives	5.462	Methionine sulfoxide	up
Bra0613	Amino acid derivatives	3.527	Phe-Phe	up
* Bra0451	Amino acid derivatives	3.329	Allysine(6-Oxo DL-Norleucine)	up
* Bra0510	Amino acid	3.095	L-alanine	up
* Bra0656	Amino acid derivatives	2.534	N-(3-Indolylacetyl)-L-alanine	up
* Bra0515	Amino acid derivatives	2.247	3-Hydroxy-3-methylpentane-1,5-dioic acid	up
Bra0031	Amino acids	0.320	L-alanine	down
* Bra0431	Amino acids	0.407	L-(−)-Cystine	down
Bra0011	Amino acids	0.469	L-(+)-Lysine	down
Bra0497	Amino acid derivatives	0.272	S-(methyl)glutathione	down
* Bra0057	Amino acid	0.353	Aspartic acid	down
Bra0638	Amino acid derivatives	0.381	Acetyl tryptophan	down
Bra0474	Amino acid derivatives	0.418	N-Acetylaspartate	down
* Bra0560	Amino acid derivatives	0.429	N-Acetylmethionine	down
Bra0022	Amino acid derivatives	0.433	2,6-Diaminooimelic acid	down
* Bra0470	Amino acid derivatives	0.445	Glutathione oxidized	down
Bra0071	Amino acid derivatives	0.448	(5-L-Glutamyl)-L-amino acid	down
Bra0072	Amino acid derivatives	0.457	Nα-Acetyl-L-arginine	down
* Bra0553	Amino acid derivatives	0.483	Asp-phe	down
Bra0045	Amino acid derivatives	0.497	N,N-Dimethylglycine	down

**Table 4 ijms-23-05784-t004:** Fold change of plant hormones metabolites detected in the HN leaf when compared to the LN leaf, and HN root compared to the LN root, respectively. Asterisk marks indicate that the metabolite was detected both in the leaf and root.

**HN-L vs. LN-L**				
**ID**	**Class**	**Fold Change**	**Compounds**	**Type**
* Bra0177	Phytohormones	300.296	Indole carboxylic acid	up
Bra0329	Phytohormones	5.791	Methyl indole-3-acetate	up
Bra0303	Phytohormones	0.188	Indole 3-acetic acid (IAA)	down
Bra0679	Phytohormones	0.168	Gibberellin A20	down
Bra0701	Phytohormones	0.153	N-[(−)-Jasmonoyl]-(L)-Isoleucine (JA-L-Ile)	down
**HN-R vs. LN-R**				
**ID**	**Class**	**Fold Change**	**Compounds**	**Type**
* Bra0177	Phytohormones	0.057	Indole carboxylic acid	down
Bra0156	Phytohormones	0.418	trans-zeatin 9-O-glucoside	down
Bra0529	Phytohormones	0.431	Salicylic acid O-glucoside	down

## Data Availability

Not applicable.

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
