# Peer review of "Metabolic Profiles Reveal Changes in the Leaves and Roots of Rapeseed (Brassica napus L.) Seedlings under Nitrogen Deficiency"

_ijms, 2022, doi:10.3390/ijms23105784_

Round 1

Reviewer 1 Report

The aim of this work was to analyze the phenotypic changes of rapeseed under normal-nitrogen (HN) and nitrogen-deficiency (LN) conditions. The Authors also performed untargeted metabolomics studies to identify and recognize the changes in the metabolites in crop plants under LN conditions. The obtained results are satisfactory because they showed that N deficiency severely inhibited rapeseed shoot growth and promoted rapeseed root architectural changes under LN conditions. Xinjie Shen et al. identified numerous metabolites accumulated in the leaves and roots, both in HN and LN conditions. Which were involved in four primary metabolic pathways. Moreover, the Authors found that plant hormones have distinct accumulation patterns in rapeseed.

Overall, this work is interesting and carefully performed. The Introduction contains significant information to justify taking up research. The studies are well-conducted, -organized, and clearly described. The Materials and Methods section provides enough details. The manuscript and data are sufficiently sound to support the conclusions. Nevertheless, I would suggest introducing a Conclusions chapter that would summarize the most relevant results obtained.

My some minor comments:

  • Line 53: Arabidopsis thaliana should be in italics (the same in other places).
  • Line 55: ‘a’ after metabolites is probably unnecessary here.
  • Line 343: Delete ‘2.8 LN Treatment’.
  • Figure 3 A and B: Replace ‘unchange’ with ‘unchanged’.
  • Lines 480, 525: What do you mean by ‘biological triplicate’?
  • Line 426: The explanation of the abbreviation for ROS should already be found in line 71. Explain also the abbreviations for RD, PPCA, and KEGG in the text when you first use them. Not all abbreviations used are depicted in the list of Abbreviations.
  • Consider whether it would be possible to combine the information from Tables 1 and 2 in one table. The same for the Tables 4 and 5.

Author Response

    Thank you for your arduous work and instructive advice. Special thanks to you for your good comments. Please see the attachment.

Reviewer 2 Report

The authors investigated the metabolic profiles in the leaves and roots of rapeseed (Brassica napus L.) seedlings under nitrogen deficiency, but this manuscript, in particular discussion and methods, should be revised further. In discussion, the authors repeatedly presented the results only with little biological implications. The most important defect is missing of the new findings from this work. Also, they did not describe how they analyzed their data in Methods.

Line 108: Need to present the full word prior to using an abbreviated form of RD.

Figure 1. Phenotypic and physiological characteristics; No physiological data, so delete “physiological” from the legend.

Figure 2. PPCA; typo change “PPCA” into “PCA”.

Line 128. Two periods (..). Delete one of them.

Lines 145 – 152: increased and decreased metabolites in WHICH condition?

Figure 3. In volcano plot, in general a y-axis is represented by -log10(p-value). The author should explain why they used variable importance in projection (VIP) instead of p-values.

Lines 180 – 181: No explanation for metabolites accumulated in leaf tissues

Figure 4. A and B are not readable. Need to present images with higher resolution. No explanation for Figure 4D in the legend. What is the q value?

Lines 196 – 200: this part belongs to Introduction, not Results.

Table 1. Need to specify “up” or “down” in WHICH condition.

Line 216: “common” should be used with caution since the authors only investigate the N deficiency.

Section 2.6 : Cysteine showed the greatest change among amino acids and their derivative, but why the authors chose “aspartic acid” instead?

Lines 242 – Lin 247: This should belong to Discussion section.

Lines 250 – 342: Line numbering problem

Line 343. Typo

Figure 5. Same comments for VIP

Line 380. There is no Table 7. No description on gibberellic acid in Results.

Discussion: repetition of results, little biological implications

Materials and methods

4.2 Phenotypic and Physiology characterization

- No physiological data

- Need more description how phenotypic data were measured

- No information on how the obtained data (both phenotypic and metabolite data) were analyzed ( no statistical analysis)

- No description on how the authors performed cluster analysis.

- No description on KEGG analysis

Author Response

    Thank you for your arduous work and instructive advice. Special thanks to you for your good comments. Please see the attachment

Round 2

Reviewer 2 Report

The authors revised the manuscript according to the comments properly. However, I still have a concern for Figure 4 since the letters on Figure 4A and 4B are not readable. Please use a better resolution for them. Also, conclusions should be presented right after discussions.
